# Degradation Mechanism of Pressure-Assisted Sintered Silver by Thermal Shock Test

**Keisuke Wakamoto** [1,2,*]**, Takukazu Otsuka** [1]**, Ken Nakahara** [1] **and Takahiro Namazu** [2]

[1] ROHM Co., Ltd., Kyoto 615-8585, Japan; takukazu.otsuka@dsn.rohm.co.jp (T.O.);
   ken.nakahara@dsn.rohm.co.jp (K.N.)
[2] Faculty of Engineering, Uzumasa, Kyoto University of Advanced Science, Kyoto 615-8577, Japan;
   namazu.takahiro@kuas.ac.jp
[*] Correspondence: keisuke.wakamoto@dsn.rohm.co.jp

**Abstract:** This paper investigates the degradation mechanism of pressure-sintered silver (s-Ag) film for silicon carbide (SiC) chip assembly with a 2-millimeter-thick copper substrate by means of thermal shock test (TST). Two different types of silver paste, nano-sized silver paste (NP) and nano-micron-sized paste (NMP), were used to sinter the silver film at 300 °C under a pressure of 60 MPa. The mean porosity ($p$) of the NP and MNP s-Ag films was 2.4% and 8%, respectively. The pore shape of the NP s-Ag was almost spherical, whereas the NMP s-Ag had an irregular shape resembling a peanut shell. After performing the TST at temperatures ranging from −40 to 150 °C, the scanning acoustic tomography (SAT) results suggested that delamination occurs from the edge of the assembly, and the delamination of the NMP s-Ag assembly was faster than that of the NM s-Ag assembly. The NMP s-Ag assembly showed a random delamination, indicating that the delamination speed varies from place to place. The difference in fracture mechanism is discussed based on cross-sectional scanning electron microscope (SEM) observation results after TST and plastic strain distribution results estimated by finite element analysis (FEA) considering pore configuration.

**Keywords:** sintered silver; porosity; thermal mechanical reliability design; fracture mechanism; low thermal impedance packaging

## 1. Introduction

Recently, reducing $CO_2$ emissions has become one of the most pressing issues worldwide for safe ecology. In the automobile industry, the replacement of fuel vehicles with electric vehicles (EVs) is progressing. EVs demand high power density systems to miniaturize the car body and expand interior living spaces. Power modules with high power density and that are small have been one of the main trends in EV power conversion [1]. To downsize power modules, wide bandgap (WBG) semiconductor devices, such as SiC and GaN, contribute significantly because WBG devices possess excellent performance of low resistance loss and switching loss [2]. To draw out device properties and apply them for power modules, new thermal management is essential, because the increase in device thermal density with chip size reduction becomes a more decisive problem than the reduction in heat generation loss.

Therefore, die attach materials with high thermal conductivity and heat spread within power modules are required. As a die attach material, sintered silver (s-Ag) has attracted many researchers [3–6]; the thermal conductivity has been reported to exceed 200 W(m/K) in the case of using pressured-assisted s-Ag [7,8], a value that broke the conventional thermal conductivity limit with around 60 W(m/K) of solder material [9–11]. As for the heat spreader, copper substrates jointed with a ceramic plate, called direct bonded copper (DBC) substrates, have been frequently used for power modules. However, the typical thickness of copper in a DBC substrate has been limited to around 0.5 mm due to production difficulties [12]. Hence, a substrate sandwich structure called a double-sided

cooling package has also been developed to expand the cooling area [13–16] to cover a small area of cooling in a single-sided DBC assembly. However, many difficulties in manufacturing double-sided cooling structures still remain. Alternatively, a substrate based on a thick lead frame with a thin-film insulation layer has been developed to provide enough heat spread even in single-sided cooling [17,18]. The copper thickness is not limited by this structure. Therefore, the combination of the s-Ag die attach material with a thick copper substrate assembly structure is expected to be one of the most desirable solutions for the low thermal impedance of WBG device packaging. Hence, this paper focuses on this structure.

However, for real usage of power modules, the thermal shock test (TST), which reflects the real operation mode, has to be passed. In the TST, power modules are set in a heat chamber. The chamber temperature has commonly been set from −40 to 150 °C or −40 to 125 °C for a duration of 30 min to 2 h in the field of the automobile industry [19,20]. Figure 1 shows the TST design issue overview. During the TST, the joint area is degraded by repeated thermo-mechanical stress induced in the die layer caused by the coefficient difference between each material. In a case with a thick copper assembly, for sufficient heat spreading to be generated at the chip, highly repeated thermo-mechanical stress is induced in the joint layer during the TST. As a result, degradation in the die layer causes shrinkage of the thermal pass area, which causes the total thermal impedance to be worse after the TST. The bonded area ratio has generally been set to above 80% after the TST [19,20]. Hence, degradation control during the TST plays a critical role for thermal designing in packages. However, in s-Ag joint assemblies, TST designing is still unsolved. One of the difficulties is the s-Ag material characteristic that inevitably includes micropores as a stress concentration area. As for the essential steps for understanding the degradation mechanism during TST, many researchers have investigated micropores' dependence on s-Ag tensile mechanical properties [20–28]. Particularly, authors focused on micropores' dependence on its tensile mechanical properties by applying changed process pressure from 5 to 60 MPa with nano-sized silver paste (NP) as a function of the s-Ag porosity rate ($p$) at room temperature [24]. Based on the results, s-Ag of a pressure above 30 MPa with around 5% $p$ shows good mechanical properties. Furthermore, pressure-assisted s-Ag temperature dependence on mechanical properties has also been investigated corresponding with the TST temperature circumstance [29,30]. The s-Ag mechanical property changed from brittle to ductile at a temperature of around 100 °C. Furthermore, our team reported that higher temperatures in the TST (30 to 150 °C) led to faster deterioration of s-Ag than the low temperature range (−60 to 60 °C) by using a 60 MPa pressure-assisted s-Ag die attach SiC chip assembly. The results indicated that degradation during the TST progresses more in a higher temperature range [31].

The other TST design issue is that it is time-consuming to evaluate due to its long cycle period time per cycle (30 min), as well as the temperature increasing rate limitation (around 2 °C/s). The evaluation time lasts around two or three months for power module production in general. To accelerate TST evaluation, some researchers have tried to use a liquid-to-liquid evaluation chamber to alternate the acceleration test of TST [32–34] because a liquid-to-liquid chamber can produce high temperature increasing rates (up to 2500 °C/s) in the TST.

The authors of [30–32] tried to perform a TST evaluation with a liquid–liquid chamber by using an AMB assembly bonded with three different $p$ levels of s-Ag: $p$ was set to 8%, 12%, and 20%. The temperature cycle pattern was set to −60 to 150 °C in each cycle. The dwell time was set to 3 min. After 3000 cycles passed, the reliability was found to increase with decreasing s-Ag $p$. The results indicated that lower $p$ of s-Ag had a good influence on TST reliability, which had good agreement with our mechanical testing results of s-Ag [27].

However, the degradation mechanism, mode, and speed during TST have not been revealed for pressure-assisted s-Ag assemblies with a thick copper substrate. Our present research investigates the degradation mechanism of a SiC chip assembly with a copper substrate 2 mm in thickness by two different types of silver paste (NP and nano-micron-sized

paste (NMP)) sintered with 60 MPa process pressure. A scanning acoustic tomography (SAT) image was taken every 100 cycles to check the bonding state after TST. The fracture evaluation is also discussed in the cross-sectional scanning electron microscope (SEM) images after TST and finite element analysis (FEA) results.

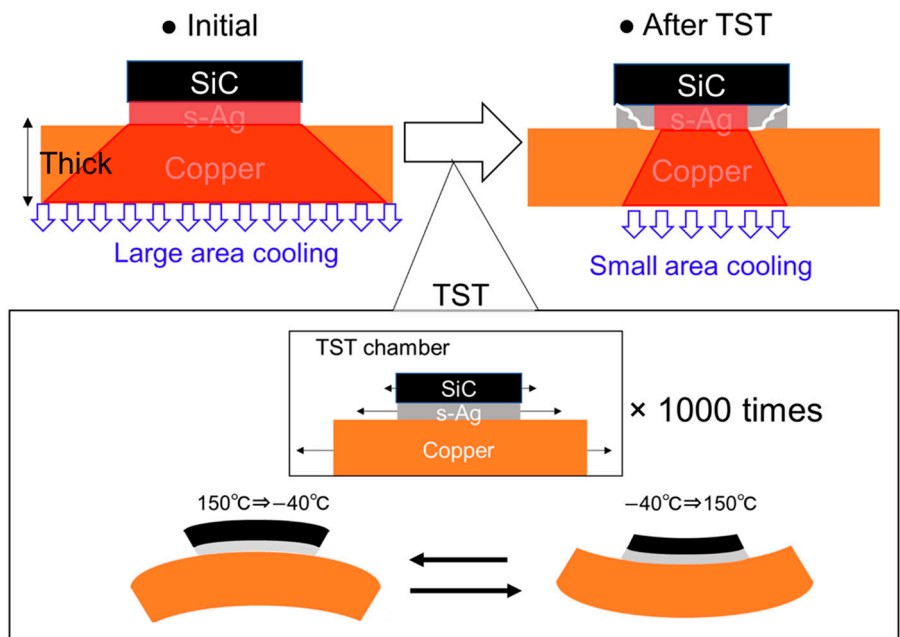

**Figure 1.** Thermal design issue overview of the TST.

## 2. Materials and Methods

Figure 2 shows an overview of the procedure for preparing a SiC chip assembly used in TST. Two different types of paste were used in this study. NM included silver with a mean diameter of 20 nm in the paste. On the contrary, NMP included silver with a mean diameter of 18, 150, and 5063 nm in paste. First, a paste with silver paste coated using an organic stabilizer was stencil-printed on the silver-electroplated copper substrate. Then, the organic solvent was evaporated at 140 °C for 60 min in NP and at 70 °C for 30 min in NMP. A dry process was selected for appropriate conditions to remove the organic solvent with each type of paste.

After that, the dried paste was sintered at 300 °C under 60 MPa pressure for 10 min via a buffer sheet that ensures homogeneous pressure distribution on the chip contact area. The lower right part in Figure 2 shows the assembly sample appearance. A 4.8-square-millimeter SiC chip, 0.35 mm thick, was bonded to 20-square-millimeter copper substrate with 2 mm thickness. The substrate surface was treated with silver plating to improve the s-Ag and substrate surface adhesive strength. The joint layer thickness was set to $55 \pm 10$ μm.

The microporous structure of s-Ag was evaluated from a 5-square-millimeter s-Ag sheet with 50 μm thickness in accordance with the SiC assembly die size. The s-Ag film manufacturing methodology was already reported in [27]. To determine the s-Ag porous state, cross-sectioning treatment was applied. First, the resin-encapsulated surface was polished with abrasive paper in a water base suspension (Struers, Labosystem). Then, buffing was carried out with a mono-crystalline diamond slurry solution. Finally, an ion milling tester (Hitachi high technology: IM4000, Tokyo, Japan) was utilized to planarize the sample surface. The planarized surface was observed using a scanning electron microscope (SEM) tester (Hitachi high technology: S4800, Tokyo, Japan). An electron acceleration voltage (Vea) of 1.5 kV and a magnification rate (Rmag) of 5000 times were used for cross-sectioned observations. Then, the obtained SEM images were binarized to illustrate the pore configuration after being imported to image software (PicMan, WaferMasters, Dublin,

CA, USA). The software can analyze each binarized pore's area and perimeter. Then, we characterized each porous state of s-Ag with its porosity rate (*pr*), pore size (*ps*), and pore shape factor (*psf*) obtained from the image analysis results, where *pr* refers to the ratio of the black area to the whole cross-sectional area, which means the degree of pore density in the cross-section layer; *ps* is each pore's area detected in the image software; and *psf* is the geometrical property that represents the irregularity of the pore shape. The parameter was described by Anselmetti et al. [35]. The *psf* was simplicity expressed as the following equation:

$$psf = \frac{P}{2\sqrt{\pi A}} \tag{1}$$

where P = pore perimeter (μm) and A = pore area (μm$^2$). For a perfect circle pore shape, *psf* should be one. When increasing the irregularity, the pore circumstance value becomes higher, which results in a higher *psf* value.

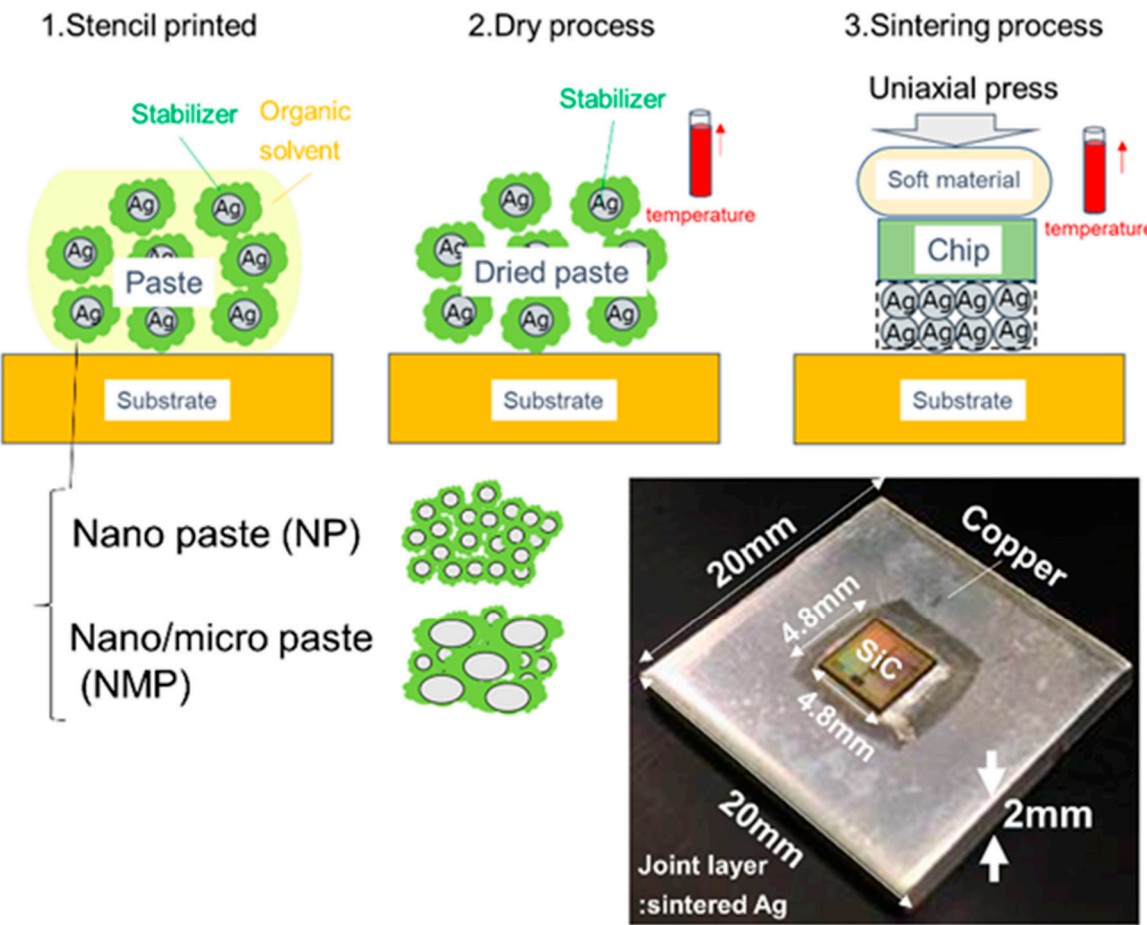

**Figure 2.** Sample fabrication steps and sample appearance.

Assembly samples were subjected to TST with the temperature fluctuating from −40 to 150 °C in each cycle. Each cycle lasted 30 min in accordance with general TST patterns as stated in the Introduction [19]. The bonded area was roughly evaluated by SAT every 100 cycles. Furthermore, in order to finely observe the SiC chip assembly after the TST cycles, cross-section observation was also performed. The cross-section was first produced by mechanical cut dicing. Then, the cross-sectioning procedure already described in this section was utilized to planarize the surface. After the cross-sectioning treatment, SEM was also performed to observe the die fracture morphology after TST completion. The electron acceleration voltage (Vea) was set at 1.5 kV. The magnification rate (R mag) was set to 100 and 5000 times for appropriate observation for characterizing die fracture morphology.

## 3. Results

### 3.1. Microporous Structure of s-Ag

Figure 3 shows representative cross-sectional SEM images of each type of s-Ag as well as the porous structure analysis results obtained from the image software. The image area for $p$ calculation was set as x = 25.4 μm and y = 16 μm. Three images were used for evaluation in each s-Ag. Image analysis with the binarizing methodology was performed with the same methodology described in previous research [27]. The two representative SEM images show that the NP s-Ag has almost spherical pores, whereas the NMP s-Ag has irregular pore shapes resembling peanut shells. According to the image analysis results, the mean $pr$, $ps$, and $psf$ values of the NP s-Ag were smaller than those of the NMP s-Ag. The mean value of $p$ for NP was 2.4%, and that of NMP was 8%. The mean value of $ps$ for NP was 0.012 μm$^2$, and for NMP was 0.046 μm$^2$. The mean value of $psf$ was 1.22 and 1.39 for NP and NMP, respectively.

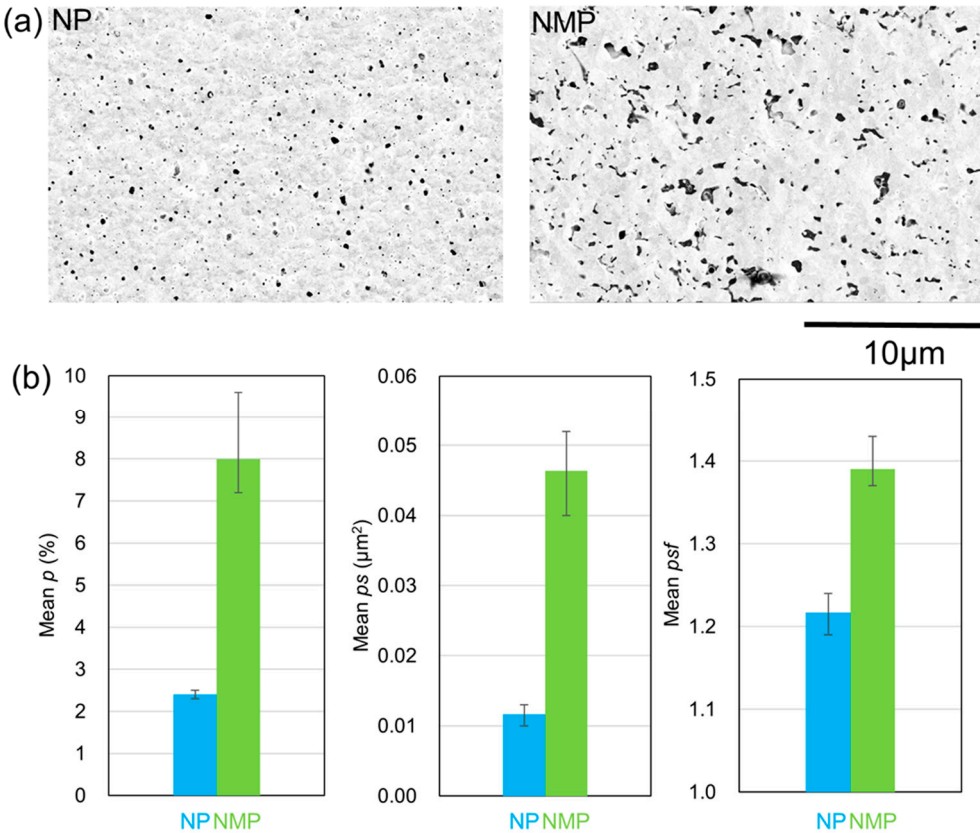

**Figure 3.** Representative pore configuration information in each type of s-Ag. (**a**) Representative cross-section SEM images of each s-Ag type. (**b**) Image analysis results of porosity rate ($p$), pore size ($ps$), and pore shape factor ($psf$). Blue colored: NP. Green colored: NMP.

### 3.2. Thermal Shock Test Results with Scanning Acoustic Tomography Images

Figure 4 depicts the SAT image results of the NP s-Ag and NMP s-Ag assemblies that passed through each TST cycle. The observed cross-section lines are indicated by red arrows in the SAT images. Sample No. 4 of the NM s-Ag assembly was chosen as a representative observation after 1000 cycles of TST. Sample No. 2 of the NMP s-Ag assembly was also chosen as a representative observation after passing 500 cycles of TST. An SAT image was taken every 100 cycles. In these images, the white area represents the unbonded area which corresponds with the reflective acoustic intensity becoming high due to the existence of air in the die. By contrast, the black-colored area represents the bonding area, which is not reflected. The upper panel shows the ideal bonded area transition during TST and the approximate bonded area ratio with each condition. Bonding degradation is

considered to occur with thermo-mechanical stress distribution. Generally, it is known that the die corner part where the thermo-mechanical stress concentrates becomes the initial die failure point, except in the case of the existence of another stress singularity point in the die layer [20,28]. In this study, the target bonded area ratio was set to 80% after 1000 cycles of TST passed, which is a conventional standard target as introduced before [19,20]. The results of the NP s-Ag and NMP s-Ag assemblies differ significantly. Degradation occurred from near the edge of almost all NP s-Ag assemblies after around 300 or 400 cycles of TST. Up to 500 cycles, all evaluated assemblies of NP s-Ag satisfied the target value. The NP s-Ag assemblies gradually degraded from 600 to 1000 cycles of TST. The degradation morphology difference among samples gradually enlarged as TST cycles increased. In addition, the degradation area and speed varied between NP assemblies. Hence, none of the NP s-Ag assemblies met the target. By contrast, the NMP s-Ag assemblies showed random degradation during the early stages of the TST with 100 to 200 cycles. Furthermore, the degradation speed of the NMP s-Ag assemblies was much faster than that of the NP s-Ag assemblies. All NMP s-Ag samples showed a die ratio lower than 50% after 500 cycles had passed. Thus, we stopped the evaluation at this stage with the NMP s-Ag assemblies.

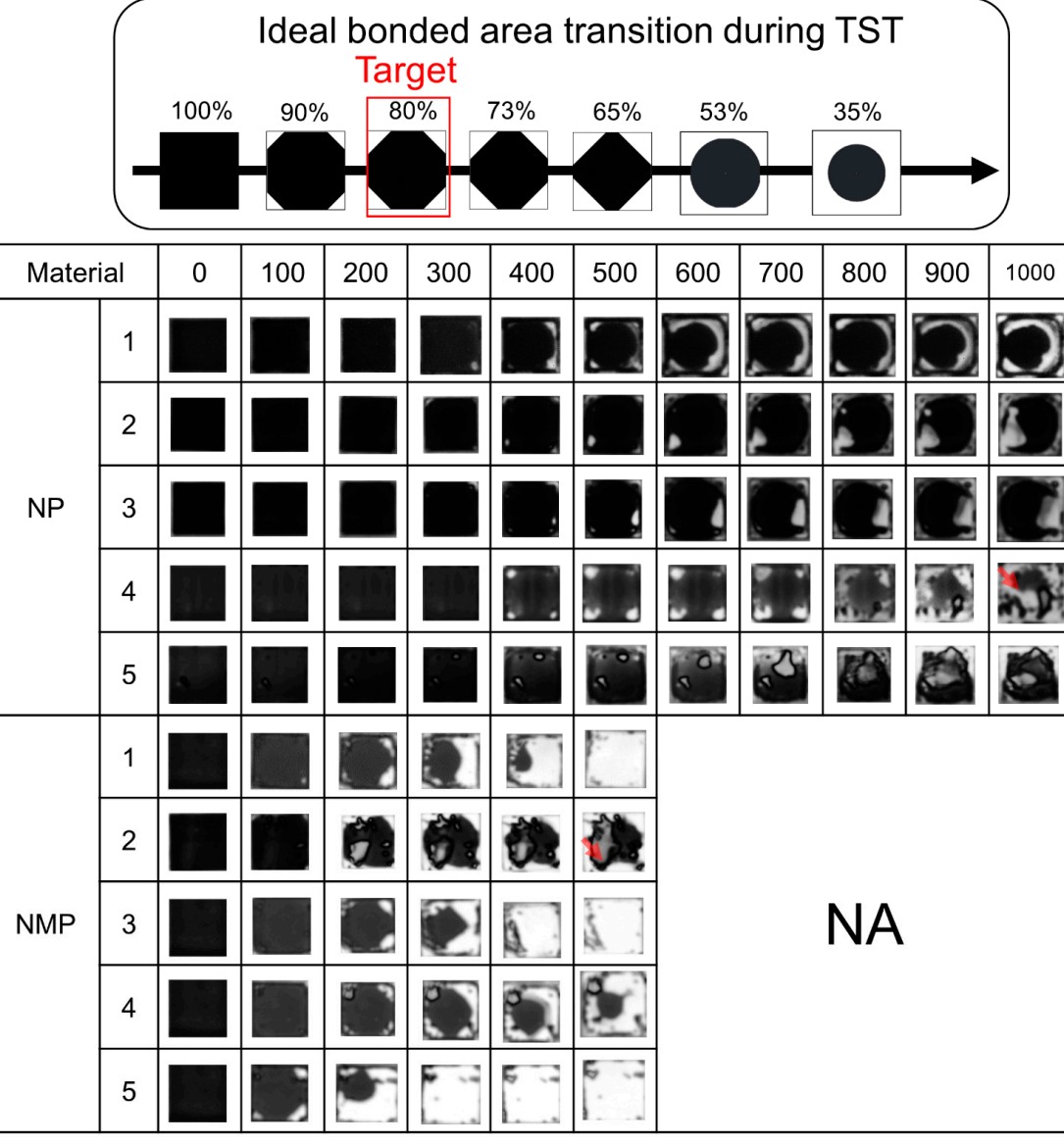

**Figure 4.** SAT images obtained after each number of cycles from 0 to 1000 (Upper image represents ideal bonded area transition during TST). Red arrows: The observed cross sectioned line.

Figure 5 illustrates the results of Figure 4 in graphs, showing the image analysis results of the bonding area ratio and the bonded area centroid shifted length from the center as functions of TST cycles. The centroid shifted length indicates the degradation non-uniformity. In the NP assemblies, the mean average die ratio was kept at 80% up to the completion of 600 cycles of TST. However, dispersion of the bonded ratio gradually increased from 500 to 1000 cycles. After 1000 cycles passed, the bonded area ratio ranged between 21% and 73% among samples. The centroid shifted value also increased from 500 cycles. This means that the bonding degradation started to be non-uniform at 500 cycles, and then the non-uniformity enlarged with increasing TST cycles. By contrast, in NMP s-Ag assemblies, even after 200 cycles, the bonded area ratio ranged between 32% and 93%. Furthermore, the mean bonded area ratio rapidly decreased to 9% after 500 cycles passed. The centroid shift length of NMP s-Ag was extremely large even after 200 TST cycles compared to that of NP s-Ag at 1000 TST cycles. Therefore, the NMP s-Ag assembly showed non-uniform fracture at around 200 cycles of TST.

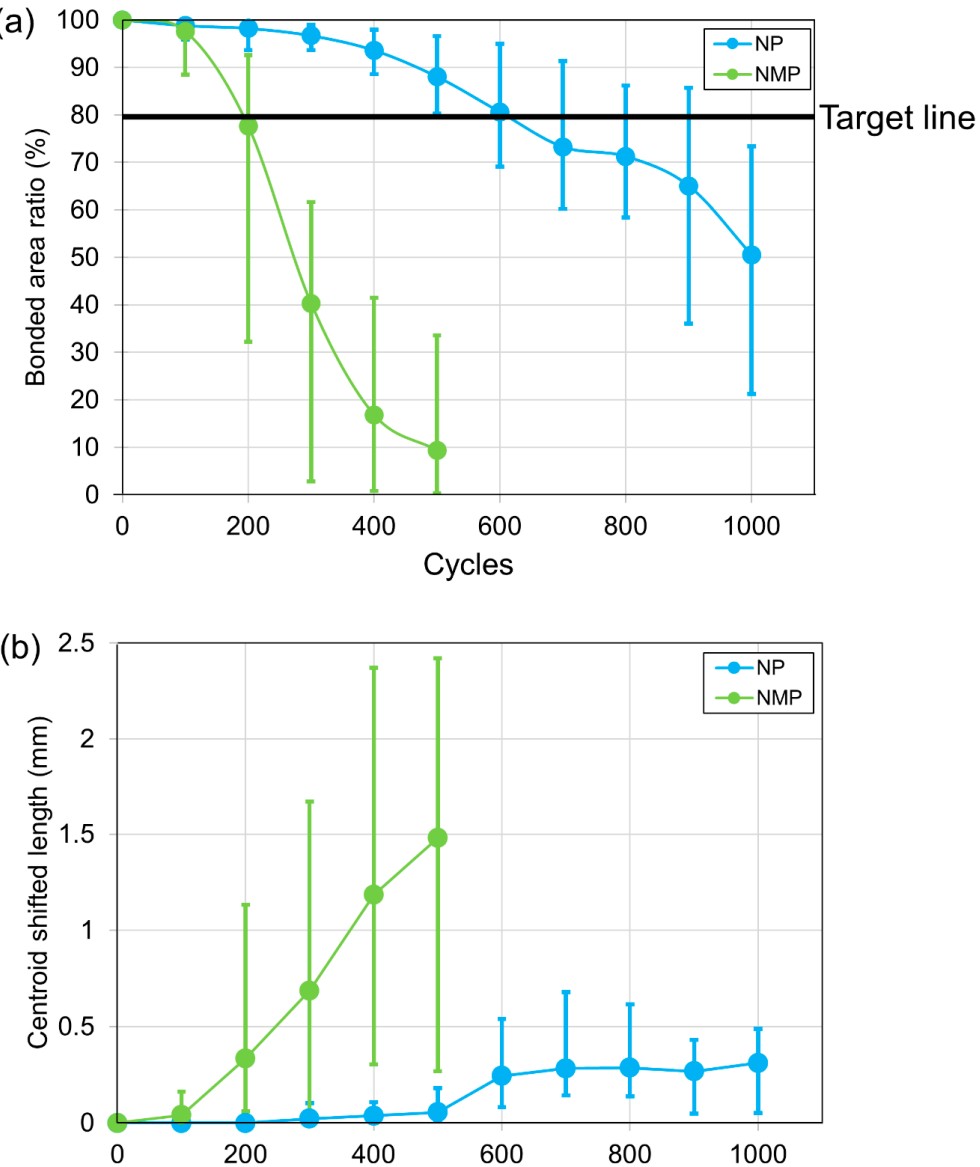

**Figure 5.** SAT image analysis results of NP and NMP s-Ag assemblies after each TST cycle. (**a**) Cycle versus bonded area ratio. (**b**) Cycle versus centroid shifted length.

## 4. Discussion

### *4.1. Cross-Section SEM Images after TST*

To discuss the fracture mode difference, cross-sectioned SEM images of each type of s-Ag assembly after TST are illustrated in Figure 6. The cross-sectioned area is illustrated in Figure 4 as red arrows. From each overall cross-sectioned SEM image, the fracture mode is divided into two groups. One is the edge crack mode that represents low *p* area, represented by the blue colored lines. In this area, the crack is almost continuous and mainly located on the copper substrate in a straight line. The crack width is larger in the NMP s-Ag assembly. No other significant damage area can be seen in this area. The images inside the blue squares are an enlarged view of the representative SEM images of this area. The porous state of s-Ag was not so much changed from the initial state of sintering. Detailed quantitative discussion will be stated in the following section. The other fracture mode is the void growth mode with large *p* area, represented by the red colored lines in the SEM images. In this area, not only a crack but also locally void growth can be seen. The crack path direction changed to an upward diagonal direction, but it was not a controllable and discontinuous crack. The SEM images in the red squares show the representative pore growth area. In comparison with the blue squared area, the *p* drastically increases. The fracture mechanism of void growth is considered to differ from that of edge cracking. The observed cracking modes in the present work also showed differences in comparison with other studies mentioned in Section 1 [30–32]. Vertical cracks in the die layer seemed to be dominant in the cross-section SEM observation results after accelerated TST in the liquid–liquid chamber. *p*, *ps*, and *psf* value differences did not seem to be observed in the bond line. On the other hand, s-Ag mechanical characteristics are known to change to a stiffer form with increasing strain speed above the temperature of 100 °C [20,28]. Therefore, the s-Ag mechanical characteristics would always show a stiffer property with increased strain speed induced by high-speed temperature change in the liquid–liquid chamber. Hence, cracks in the vertical direction can be ascribed to the principal mechanical tensile stress induced in the longitudinal direction of the substrate. Therefore, the void growth area in this work is mainly attributed to the microstructural aging behavior caused by slow strain speed during TST.

Figure 7 shows the image analysis results of *p*, *ps*, and *psf* with each state. Three images were analyzed for evaluation. The crack edge area shows a lower p in comparison with the initial state of s-Ag. The mean *p* of NP s-Ag decreased from about 2.4% to 0.9%, and the NMP s-Ag mean *p* decreased from about 8% to 2.2%. By contrast, in the area where pore growth increased after TST, the mean *p* of NP s-Ag increased from about 2.4% to 12%, and that of NMP s-Ag increased from about 8% to 20.9%. The increase in *p* has a negative effect on mechanical strength [27], which makes the material more susceptible to fracture in this region. Regarding mean *ps*, both types of s-Ag increased after TST regardless of the fracture mode. In particular, the material degradation area rapidly increased from the initial state. As for *psf*, in the NP s-Ag, it increased in the material degradation area after TST. The *psf* of NMP showed a high mean value of 1.39 in the initial state and showed no significant difference after TST. Based on the cross-sectioned image results, the pore growth area after TST may cause uncertain fracture, and suppressing the phenomenon plays a critical role for thermal design during TST.

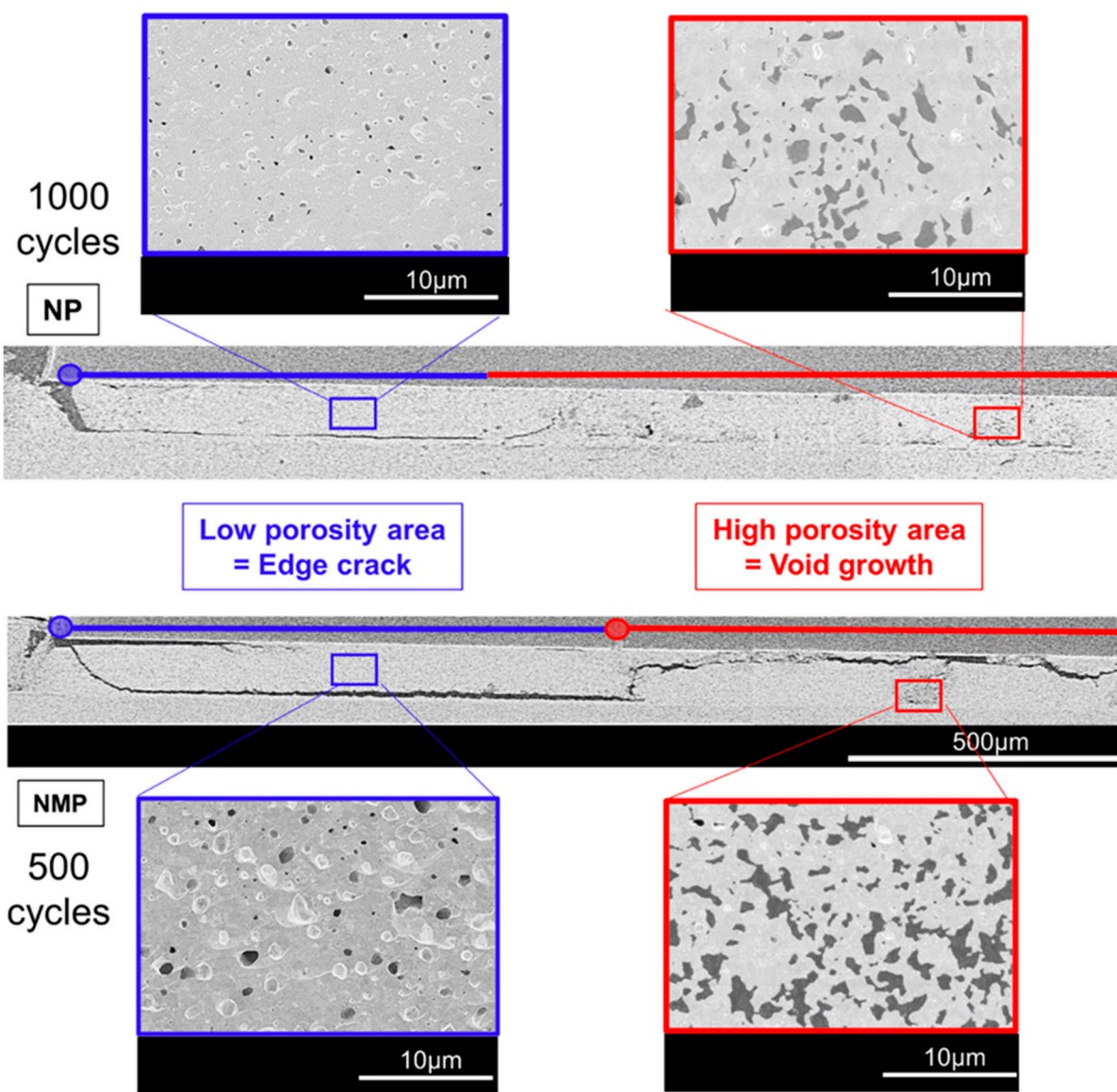

**Figure 6.** Fracture cross-section observation results of NP and NMP die area. Blue lines show low porosity rate (*p*) area; red lines show high porosity rate (*p*) area. (NP: after 1000 cycles of TST; NMP: after 500 cycles of TST).

### 4.2. Strain Distribution by Finite Element Analysis

In the next stage, FEA was conducted for plastic strain distribution around each s-Ag porous state to consider the pore configuration effects on fracture. Figure 8 depicts the FEM modeling overview and Mises plastic strain distribution in the die layer during TST. The simulation model takes into consideration each s-Ag assembly's *p*, pore location, and pore figure, as can be seen from the enlarged view of the model. In this simulation model, a crack was also introduced into the die layer, which corresponds with edge crack generation. These image layers were fabricated via image processing software (Simpleware, Synopsys) to import the initial cross-section SEM image of each s-Ag. Two-dimensional symmetry configuration was selected to reduce the total calculation time. Then, the model was also meshed with triangle shape in Simpleware. The boundary condition was set to symmetry in thick black line. Point binding was also set at the bottom of the symmetry line as represented in Figure 8. Furthermore, the cyclic temperature was set from −40 to 150 °C. The holding time of the TST was not considered in this study. The input material characteristic was elastic for SiC and copper. In this study, the bilinear elastic–plastic

mechanical property of s-Ag was employed. The modeled bilinear line refers to s-Ag's temperature dependence in the stress–strain curve that was measured from a previous study's results [30]. Each input value is shown in Table 1. FEM calculation was conducted in ANSYS 19.0. In the NP and NMP s-Ag plastic Mises strain distribution results, Mises strain was concentrated at the tip of the induced crack in both types of s-Ag die. However, in the NMP s-Ag die layer, which has larger and irregular pore shapes, it was concentrated not only at the crack tip but also internally around the pore shapes. In the NP s-Ag die layer, Mises plastic strain was also concentrated around the pores, but the area was small compared with that in the NMP s-Ag due to the small pore shape. In this regard, the NMP s-Ag has more candidate crack initiation points inside than the NM s-Ag does. Therefore, crack initiation in an NMP assembly possibly occurs not only at the edge corner where the stress concentration is inherently high but also in the internal area.

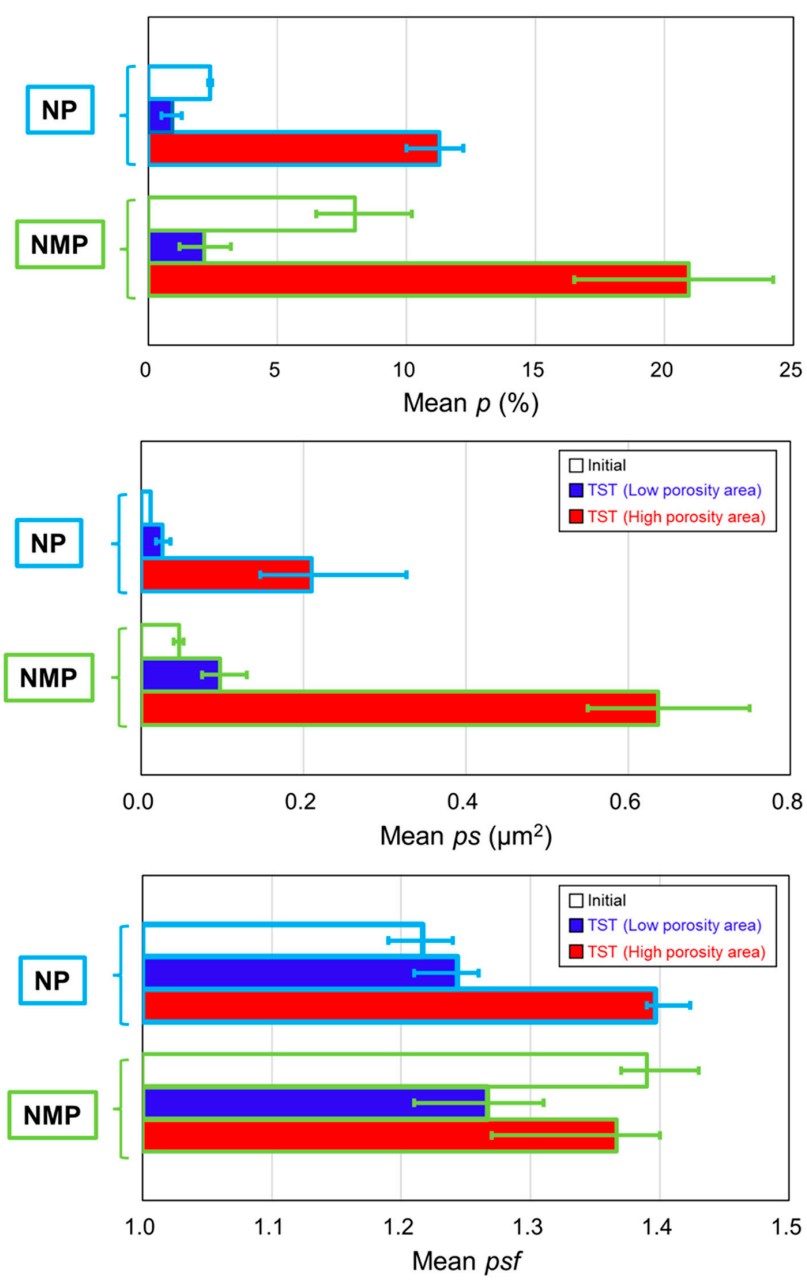

**Figure 7.** Image analysis results of porosity rate (*p*), pore size (*ps*), and pore shape factor (*psf*) with each state.

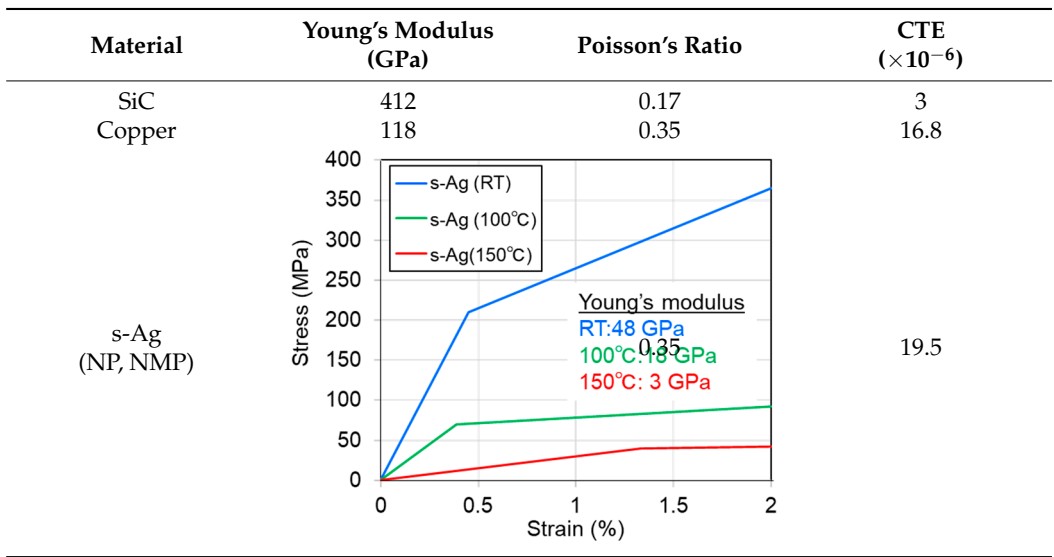

**Figure 8.** FEM overview and obtained calculation results of Mises plastic strain distribution in NP and NMP die layers.

**Table 1.** Material properties used for FEM calculations.

| Material | Young's Modulus (GPa) | Poisson's Ratio | CTE ($\times 10^{-6}$) |
|---|---|---|---|
| SiC | 412 | 0.17 | 3 |
| Copper | 118 | 0.35 | 16.8 |
| s-Ag (NP, NMP) | | 0.35 | 19.5 |

### 4.3. Fracture Mechanism

Figure 9 shows fracture mechanism images of the NP and NMP s-Ag die layers during TST, combining all obtained results. For the initial state, the pore distribution differs between. the NP and NMP s-Ag die layers due to the different sizes of the nano-pastes. Here, the NMP s-Ag die layer includes larges pore of an irregular shape. A higher plastic strain intensity is provided around the irregular pore shapes, which speeds up pore growth. As a result, an unpredictable inner fracture is caused in the early stage of the TST, which causes the bonding degradation mode to destabilize. By contrast, accumulated plastic strain growth is mild in the case of an NP s-Ag die assembly due to its small spherical pore configuration, where fracture naturally occurs at the corner of the edge part first. However, even in the NP s-Ag assembly, pore growth increases as enough TST cycles pass. When the pores grow, the fracture morphology becomes destabilized. As a result, the fracture mode in the bonding layer is scattered. In any case, it should be noted that reducing pore growth in the bonding layer is a critical way for designing TST.

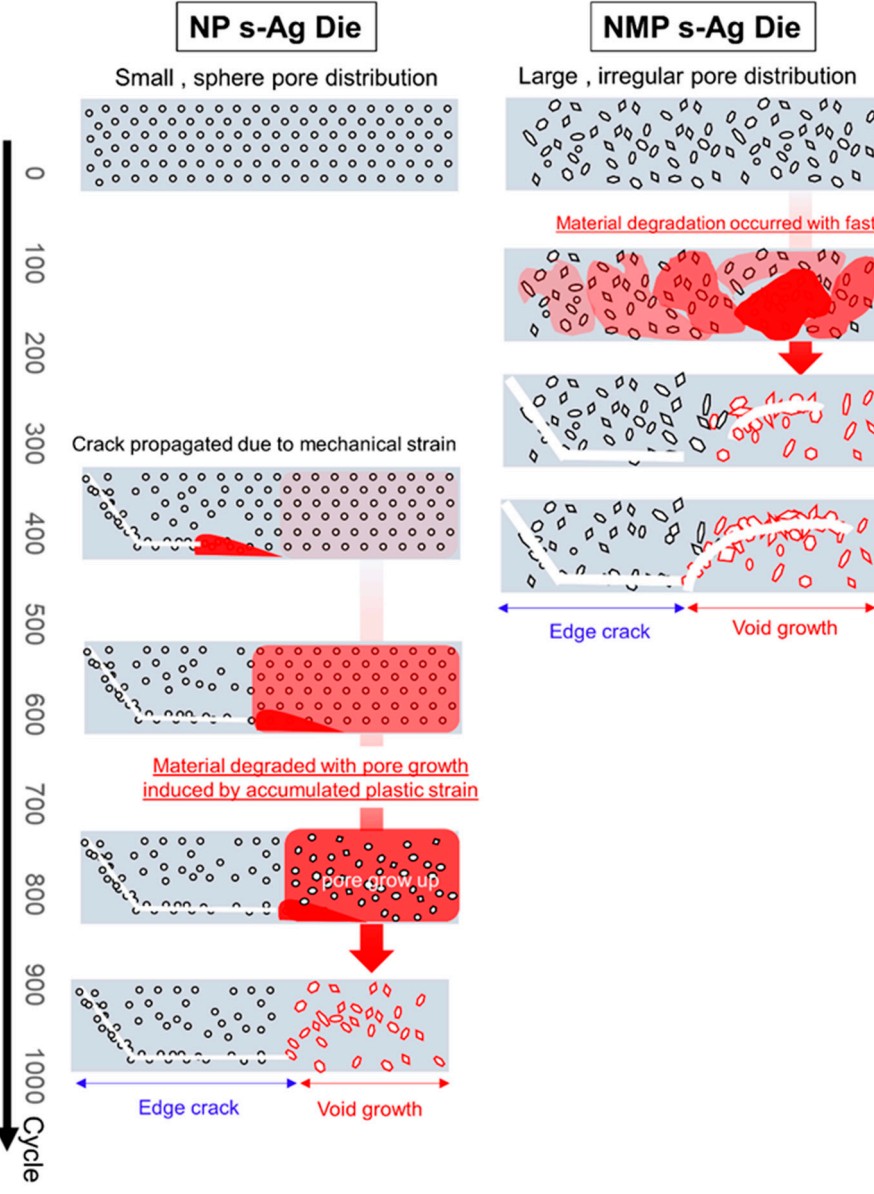

**Figure 9.** Fracture mechanism images of NP and NMP s-Ag die during TST.

## 5. Conclusions

We performed a TST assessment of two types of pressure-assisted s-Ag pastes (NP, NMP) bonded with 2-millimeter-thick copper substrates. The target bonded area ratio was set to 80% after 1000 cycles of TST passed. The TST temperature pattern was set to $-40\,°C/150\,°C$. The mean $p$ was 2.4% and 8%, the mean $ps$ was 0.012 and 0.046 $\mu m^2$, and the mean $psf$ was 1.22 and 1.39 for the NP and NMP s-Ag films, respectively. The NMP s-Ag assemblies reached the target bonded value after 200 cycles of TST. The fracture area of the NMP assemblies showed a random delamination pattern indicating that degradation speed varied from place to place. By contrast, NM s-Ag assemblies' delamination speed was much slower than that of the NMP s-Ag assemblies. The degradation area progressed along with the stress distribution up to around 400 cycles. However, the degradation speed and area varied for each assembly after 500 cycles. NM s-Ag assemblies held the target bonded value up to 700 cycles of TST. However, this was not the case with the target of 1000 cycles. In the cross-sectioned SEM images after TST, the fracture mode was divided into two groups with pore growth or not. The first was edge crack mode, which first progressed with an incline angle against the substrate, then the crack proceeded along the substrate. The second was void growth mode, which was observed inside the bond line. The $p$ and $ps$ values with both types of s-Ag in the pore growth area were drastically increased in comparison with the initial state of s-Ag. Among them, the pore growth area creates difficulties in designing a TST. As the fracture mode was unpredictable, the fracture point in the pore growth area was considered to be stress singularity point in the die line. The pore growth speed was also faster in the NMP s-Ag in comparison with the NM s-Ag. The cause was considered to come from the initial pore state, which led to the high level of stress-concentrated area in the die layer. In this regard, product makers should provide controlled nano-sized pastes for consumers to reduce stress intensity points induced in the bond line. Stress relief induced by the die layer should be provided to reach the bonded target after TST. In this regard, use of a thicker die is one of the possible effective solutions. Furthermore, the mechanical durability of s-Ag at higher temperatures is also important for designing the material degradation speed, as pore growth is attributed to micropore structural aging induced by stress, temperature, and time. For example, obtaining the creep mechanical property of s-Ag is needed to understand s-Ag aging behavior. Although not achieved in this study, it is necessary to find the correlation between the amount of thermal stress during TST and the mechanical property results of s-Ag, which will allow to establish a valid TST evaluation method.

**Author Contributions:** Conceptualization, K.W., T.O., K.N. and T.N.; methodology, K.W.; formal analysis, K.W.; writing—original draft preparation, K.W.; writing—review and editing, K.W. and T.N.; project administration, K.N. and T.N. All authors have read and agreed to the published version of the manuscript.

**Funding:** This research received no external funding.

**Institutional Review Board Statement:** Not applicable.

**Informed Consent Statement:** Not applicable.

**Data Availability Statement:** Not applicable.

**Acknowledgments:** I wish to thank Miyazaki (JSOL Company) for timely advice in developing the FEM mesh methodology.

**Conflicts of Interest:** The authors declare no conflict of interest.

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
