# Peer review of "Degradation Mechanism of Pressure-Assisted Sintered Silver by Thermal Shock Test"

_energies, doi:10.3390/en14175532_

Round 1

Reviewer 1 Report

The work requires taking into account some minor remarks:

  1. I suggest supplementing the list of literature with at least 5 works from the period 2016 plus.
  1. Literature 14 position - the year is missing.
  1. The Discussion section should be supplemented with references to the works of other authors.
  1. Conclusions from the work should be detailed.

Reviewer 2 Report

In this study, the degradation mechanism of pressure sintered silver (s-Ag) film for SiC chip assembly with 2mm-thick copper substrate by means of thermal shocked test. The results show that after TST, NMP s-Ag assembly reached target bonded value after 200 cycles of TST passed. By contrast, NM s-Ag assembly hold taget bonded value up to 700 cycles of TST. The difference in fracture mechanism is discussed based on SEM after TST and plastic strain distribution results estimated by FEA taking into pore configuration. The results are benefit for this field. Reading and understanding this manuscript are fine in the current form except for a few grammatical errors. However, I suggest the English polishment is needed.

Ex.:

1.The term “TST” , “NP” …etc should defined again in the introduction part first time.

  1. line 37, ….copper substrate jointed with celamic plate called DBC (Direct Bonded Copper) substrate has been frequently used …. The priority should be Direct Bonded Copper (DBC).

…etc  Please check.

Reviewer 3 Report

The authors analyze the degradation mechanisms of sintered silver film (with thick copper substrates), together with 2 different types of paste. The main test used was the thermal shocked test (TST).

The article is very important and very well highlighted by the tests performed and the discussions on the results.

Congratulations to the authors for their work and I would like to make one recommendation: a practical conclusion should be drawn for producers of such NP or NMP, which show ways to improve these types of silver paste.

Reviewer 4 Report

The authors proposed a degradation mechanism of pressure assisted sintered silver by thermal shocked test. They investigated a degradation mechanism of pressure sintered silver (s-Ag) film for SiC chip assembly with 2mm-thick copper substrate by means of thermal shocked test. They claimed that delamination occurs from the edge of the assembly, and the delamination of NMP s-Ag assembly is faster than that of NM s-Ag assembly. The also claimed that the NMP s-Ag assembly shows a random delamination indicating that delamination speed varies from place to place.

Although the effort and time spent in doing this research work and in writing this paper by the authors are appreciated, the following review points should be considered and should be addressed by the authors: 

1.

The linguistic rules of technical English writing have not been observed whilst writing this manuscript. For instance: on page 2  in section 1. Introduction” the authors wrote: “However, degradation mechanism, mode, and speed during TST has not been revealed among pressure assisted s-Ag 66 assembly with thick copper substrate”. This should be corrected and rewritten as: “However, degradation mechanism, mode, and speed during TST have not been revealed among pressure assisted s-Ag 66 assembly with thick copper substrate”.

Thus, the manuscript should be revised thoroughly such that the linguistic rules of technical English writing are observed. 

2.

The introduction does not provide a sufficient background.  For instance, several papers addressed this topic and should be cited for comparison with the results of the present paper. 

[Marco Santopa, Sebastiano Russo, Marco Torrisi, Marco Renna, Alessandro Sittay, Michele Calabretta,

 "Fast transient thermomechanical stress to set apressure-assisted sintering process,"

 Conference Paper, 2019  (DOI: 10.1109/PRIME.2019.8787819)]

[Michele Calabretta, Alessandro Sitta, Salvatore Massimo Oliveri, Gaetano Sequenzia,  "Silver Sintering for Silicon Carbide Die Attach:Process Optimization and Structural Modeling," Appl. Sci. 2021, 11, 7012. (https://doi.org/10.3390/app11157012)]

3.

On page 12 in section “4.3. Fracture Mechanism” there is no caption for the illustrated figure ! This has to be corrected. 

4.

The authors should Compare their results with other relevant publications that are currently available. For instance, the authors should compare with the following publications their research results:

* [Marco Santopa, Sebastiano Russo, Marco Torrisi, Marco Renna, Alessandro Sittay, Michele Calabretta,

 "Fast transient thermomechanical stress to set apressure-assisted sintering process,"

 Conference Paper, 2019  (DOI: 10.1109/PRIME.2019.8787819)]

* [Michele Calabretta, Alessandro Sitta, Salvatore Massimo Oliveri, Gaetano Sequenzia,  "Silver Sintering for Silicon Carbide Die Attach:Process Optimization and Structural Modeling," Appl. Sci. 2021, 11, 7012. (https://doi.org/10.3390/app11157012)]

5.

No mention of the limitations into this research paper.   Thus, the authors should address the limitations into this research.
